# Untying the Text: Organizational Prosociality and Kindness

**DOI:** 10.3390/bs13020186

**Published:** 2023-02-18

**Authors:** Rona Hart, Dan Hart

**Affiliations:** 1School of Psychology, University of Sussex, Falmer, Brighton BN1 9RH, UK; 2Birmingham Business School, University of Birmingham, 116 Edgbaston Park Road, Birmingham B15 2TY, UK

**Keywords:** prosocial organizational behavior, kindness

## Abstract

The scholarly field of organizational prosociality is experiencing a renewed interest, yet despite its long track record, researchers still disagree on the definitions of primary concepts. Two umbrella terms, prosocial behaviors and kindness, are particularly baffling, as they are defined similarly, at times used synonymously, yet the differences between them are unclear. Consequently, the field suffers from conceptual ambiguity, which hampers its development. In this brief critical paper, we provide a review of the definitions of prosocial behavior and kindness, in an attempt to semantically untie the text, unpack the context, and discuss the subtext that underlies these concepts. Our analysis suggests that the two concepts overlap in their emphasis on dispositions and actions that aim to promote the welfare of others. However, acts of kindness and prosocial behaviors differ in actors, their target recipients and scale. Acts of kindness are performed by an individual and directed at a person or a small group, while prosocial behaviors can be performed by a person or an organization, and can be directed at a person or a group, but may also be directed at a much larger entity: an organization, community, nation, or society at large.

## 1. Introduction

In the past 10 years, we have witnessed a sharp rise in the theoretical and empirical research on prosociality, in psychology, business studies, and other disciplines [1,2]. Prosociality is conceptualized as an umbrella term that encompasses a multitude of behaviors, dispositions (traits, states, perceptions, intentions, or motivations), or processes that benefit others or focus on the welfare of others [3]. Over the years, numerous concepts have been explored in the literature that fall within the sphere of prosociality, including helping behaviors [4], kindness [5], altruism [6], perspective taking [7], empathy [8], sympathy [9], compassion [10], caring [11], social support [12], prosocial spending [13], generosity [14], donating [15], volunteering [16], prosocial personality [17], and beneficial action [18], to name a few. In parallel, within the prosocial organizational behavior literature (a subdomain of prosociality), numerous additional terms have been coined and concepts have emerged that revolve around prosocial behaviors that specifically occur in the work domain. These include, for example, Organizational Citizenship Behaviors [19], Corporate Social Responsivity [20], Environmental, Social, and Corporate Governance [21], servant leadership [22], social entrepreneurship [23], well-doing [24], public service motivation [25], corporate philanthropy [26], and others.

The abundance of concepts indicates that prosociality is a multidimensional construct that can transpire in numerous forms. Penner, Dovidio, Piliavin, and Schroeder [27] classified the research on prosociality into three investigative levels (micro, meso, and macro), each containing different forms of prosocial behaviors or dispositions:The micro level focuses on intraindividual factors and explores prosocial dispositions and tendencies. This level of analysis explores concepts such as perspective taking, compassion, empathy, prosocial motivation, or prosocial personality.The meso level of analysis is interpersonal and examines behaviors or actions that occur within actor–recipient dyads, considering their context. Research that adopts this level of analysis focuses on concrete behaviors such as helping, caring, supporting, cooperation, altruism, generosity, and heroism.The macro level investigates prosocial actions that occur in larger contexts such as groups, communities, or organizations. These types of prosocial behaviors can occur through behaviors such as volunteering, donating, social activism, organization citizenship behaviors, social entrepreneurship, or servant leadership. However, at times, they can manifest as a norm, a process, or a set of values within a society or an organization, such as corporate social responsivity or environmental, social, and corporate governance.

Bolino and Grant [2] provided another explanation for the plethora of prosocial concepts that emerged over the years, suggesting that this is partly due to the different conceptual levels that terms represent. The authors differentiated between three types of concepts:Broad umbrella terms, such as prosociality, well-doing, beneficial action, helping behaviors, or kindness.Intermediate composite concepts that encompass within them a particular set of concepts, such as Organizational Citizenship Behaviors or Corporate Social Responsibility (CSR), Environmental, Social, and Corporate Governance (ESG), or prosocial personality.Primary concepts that refer to a particular behavior or disposition, such as altruism, prosocial spending, empathy, or compassion.

Given that these concepts are nested within each other, they necessarily overlap in some senses, as well as having some distinctive features; however, these overlaps and distinctions are rarely clarified [2].

The emergence of new prosocial concepts and the rise in research publications is also indicative of a more profound change in the social responsibility agenda within organizations. For example, the Corporate Social Responsibility (CSR) and Environmental, Social, and Governance (ESG) concepts are relatively new paradigms within the business world that emerged (or redeveloped as is the case of CSR) in the past 20 years [28,29]. Both indicate a shift in organizations’, governments’, and other stakeholders’ understanding of the role that businesses can play in solving social problems, contributing to the common good, and promoting sustainable development through socially responsible actions [30]. Similarly, from employees’ perspective, the millennial generation tends to assume responsibility for environmental sustainability and hence engage more than previous generations in volunteering, social activism, and advocacy for corporate social responsibility and ethical governance in the ecological and human sides of business operations [31]. As this generation places a greater focus on personal fulfillment and meaningful work [32], they expect employers to show more humanity, care, and respect, and display more prosocial behaviors [33].

However, the evolution of the field has attracted repeated critique that the field of prosociality lacks clarity and consensus around the definitions of core concepts [1,34], including umbrella terms such as prosocial behavior, helping behaviors, and kindness. Another persistent issue is the lack of distinction between the growing number of concepts, which blurs our understanding of their features, and how these terms overlap and differ. In an early paper, Dovidio [35] (p. 363) noted that the terms prosocial behavior, helping, and altruism are often used interchangeably, and that “there is little consensus concerning how these terms should be defined or distinguished”. Two decades later, Bierhoff [36] made the same point, and went on to untangle these terms. Nevertheless, the ambiguity and indistinctiveness of these key concepts, and numerous others that emerged more recently, seem to linger. In a recent paper, Gilbert et al. [34] (p. 2259) argued that “concepts such as prosocial, altruism, helping, kindness, love, caring, concern, compassion, empathy, sympathy and benevolence are used interchangeably”. The authors then unraveled the concepts of kindness and compassion, highlighting their areas of intersection and variance. Similarly, Pfattheicher et al. [1] (p. 124), who described the field of prosociality as being “wild and untamed”, have untied the concepts of prosocial behavior and altruism, and noted that despite the long track record of 40 years of research, “disagreements and confusion about how to define prosocial behavior, and the closely related concept of altruism, are still an actuality”.

One explanation for the conceptual imprecision that afflicts the field of prosociality is that research has been multidisciplinary, therefore drawing on different conceptual traditions and disciplinary meanings that are attached to prosociality and the many terms that fit under its umbrella. Prosociality as a scientific topic has strong roots in psychology and features in several of its subdisciplines, including social psychology [37], developmental psychology [38,39], evolutionary psychology [36], personality psychology [27], positive psychology [40], and neuroscience [41]. More recently, it has been studied in other social sciences, including education [42,43], health care [44,45], law [46], media [47], IT [48], business studies [2], and others. While an interdisciplinary approach brings many benefits to a developing scientific area, issues around jargon and conceptualization can curtail that development. Pellmar and Eisenberg [49] (p. 42) made the following observation: “Scientists trained in a discipline learn to speak a specific language and adopt the analytical and methodological constructs that have accumulated in that discipline … But it can present obstacles to interdisciplinary research”. More problematically, researchers within a discipline are often unaware of discoveries that occur in other disciplines, especially if the phenomenon has a different term in the other discipline, hence at times may rediscover one another’s findings [49].

As research on prosociality advances, the need for clarity and precision in the use of terms, concepts, and definitions and their respective level of analysis is becoming more pressing, as the field suffers from conceptual ambiguity, inconsistencies, and contradictions, which hampers its development. It creates confusion as to which concept to use and which scale matches the chosen concept, and undermines researchers’ capacity to compare findings across studies [1,2].

One lingering area of ambiguity that we focus on in this paper is the association between prosociality and kindness. Kindness is an umbrella term that is considered a subdomain of prosociality. However, confusingly, similar to the definition of prosociality, it is commonly defined as behaviors intended to benefit others [5]. Furthermore, the two terms are often used interchangeably both in the psychological and management literature (see, for example, [50,51,52,53,54,55,56], and the association between the terms is rarely clarified in research papers. This raises the question of in what ways prosociality and kindness correspond and how they vary?

In this brief conceptual and critical paper, our aim is to review and unpack the definitions of prosociality and kindness and discuss the association between them, with particular attention to the ways they are conceptualized and discussed in the organizational literature. Through this semantic exercise, we seek to untie the text by elucidating each concept, distinguishing their varied contexts, and disclosing their subtext by highlighting their overlaps and distinctions.

We wish to clarify that the analysis offered below is descriptive, in the sense that it captures how these concepts are used in the most recently published literature, and although it is critical, it is not prescriptive in a sense of attempting to offer the “right” way of conceptualizing or defining these concepts.

In what follows, we first unpack the definition of prosociality as it is cited in the psychology literature and in the organizational literature. We then present a similar analysis referring to the concept of kindness. We conclude the paper by drawing a comparison between prosociality and kindness, highlighting their areas of intersection and divergence.

## 2. Prosociality/Prosocial Behavior

Within psychology, the term prosociality (often used as synonymous with prosocial behavior) was originally coined to contrast with antisocial behavior [57]. Over the years, it was defined in several ways, which may seem deceptively closely worded, yet have distinct focal points. Drawing on Penner et al.’s [27] classification cited earlier, which differentiates between micro, meso, and macro concepts, two focal points can be discerned among the numerous definitions of prosocial behavior. One focal point emphasizes intraindividual dispositions and tendencies, and is akin to Penner et al.’s [27] microlevel concepts. The other focal point that represents meso and macro concepts is interpersonal, and concentrates on actual behaviors or actions that can be directed to a person, a group, an organization, a community, or a higher social goal.

Batson and Powell [57] (p. 463) defined prosociality as a “broad range of actions intended to benefit one or more people other than oneself”. This definition encompasses a large array of behaviors, as well as tendencies and dispositions—traits, emotional states, perceptions, intentions, or motivations that are other-orientated and intended to benefit them. The subtext in this definition according to Pfattheicher et al. [1] is that the mere inclination to benefit others meets the criteria for a disposition, intention, or a behavior to be considered prosocially orientated, regardless of whether it culminates in taking action, and regardless of the outcomes, which could potentially be unproductive or even counterproductive. Another definition that follows the same line of reasoning defines prosocial behavior as voluntary behaviors intended to benefit others [58]. The term voluntary adds another layer to the intentional aspect of this definition and emphasizes that it is self-initiated by the giver, as opposed to professional helping behaviors (conducted by medical professionals, police, social workers, teachers, and many others), which are performed as part of one’s work, and therefore not considered prosocial as they lack the voluntary aspect [36].

Focusing on prosocial behaviors or actions, Eisenberg and Miller [59] (p. 92) defined prosocial behavior as “voluntary, intentional behavior that results in benefits for another”. In this definition, both intentions and the voluntary nature of the behavior are emphasized, alongside the outcomes, which result in benefiting others. The subtext that underlies this definition is that behaviors are observed and evaluated both through the motives that drive them, but more so through their consequences [1]. This raises an intriguing question regarding who can judge whether a behavior benefits others—the actor, the recipient, or others? In an attempt to address this point, Wispé [60] suggested that prosocial behaviors are those that effectively contribute to the well-being of a person or a group, are valued by society, aligned with its norms, and have positive social consequences.

An interesting point to highlight is that in the definitions cited above (particularly those that refer to behaviors) there is no indication that prosocial behaviors should be selfless in order to meet the criteria for prosociality. That is, although prosocial behaviors are judged by their outcomes for the recipient, they can indeed be driven by selfish or selfless intention, or a combination of both, and can result in benefiting the giver in addition to the receiver [61]. Within the prosociality umbrella, only the concept of altruism is characterized by behaviors that are driven by selfless intentions and having no expectations of reward or benefit [37,62].

In one of the most comprehensive definitions of prosociality presented recently, Bailey et al. [3] (p. 1) defined prosociality as an umbrella term that encompasses “a broad set of behavioral, motivational, cognitive, affective, and social processes that contribute to, and/or are focused on, the welfare of others”. This definition is more inclusive than the definitions presented earlier, since it includes behaviors, dispositions, and processes, as well as consequences and contexts. It also brings together under one umbrella term the three levels of analysis presented by Penner et al. [27] (micro, meso, macro) and their respective concepts.

However, this wider definition brings to the fore a question regarding the terms prosocial behavior and prosociality, which are often used interchangeably. We maintain that the term prosociality and prosocial behavior are not synonymous terms, and that the broad definition offered by Bailey et al. [3] is more apt for describing prosociality, while the term prosocial behavior should refer to a subset of concepts that manifest in a behavior or action, with or without the underlying subtext of being conducted voluntarily, or evaluated according to their outcomes.

## 3. Organizational Prosociality

Within the organizational and management literature, much of the earlier research seems to adopt similar definitions to those used in psychology, though recent work on organizational prosociality seems to take a slightly different approach. An example of an early conceptualization is Brief and Motowidlo’s [63] often-cited definition of prosocial organizational behavior, portraying it as a range of behaviors intended to promote the welfare of individuals, groups, or the organization. This description is akin to Batson and Powell’s [57] description of prosociality cited earlier and focuses both on dispositions and on behaviors.

A later definition offered by Schroeder and Graziano [64] (p. 255) (that emerged in the psychological literature but seems to be applied mainly in the organizational literature) defines prosocial behaviors as “any action that benefits another”. In this definition, the focus is on behaviors and outcomes. Pfattheicher et al. [1] maintained that the subtext that underlies this definition is that the focus is on actual behavior, and these are evaluated through their consequences, regardless of whether the behavior was intentional or not, or voluntarily conducted. A key issue with this definition is that the differences between helping behaviors and prosocial behaviors become blurred. The focus of this definition on behaviors alludes to all types of professional helping behaviors, that may be motivated by the fulfillment of professional obligations, and are therefore not considered discretionary. To clarify, helping behaviors are described as interpersonal interactions where people offer each other some form of support [14]. According to Bierhoff [36], some of these behaviors indeed overlap with prosocial behaviors, while other helping behaviors are not considered prosocial. The key difference between the two categories of helping behaviors is that those that are voluntary are considered prosocial behaviors, while helping behaviors that are role-prescribed are not considered prosocial. However, this nuanced distinction has not been applied within management literature. This raises the question of how do the concepts of prosocial organizational behaviors and professional helping behaviors differ? Although Schroeder and Graziano [64] made an attempt to define and unpack both concepts, they were unable to untangle them and clarify the differences between them. To complicate this point further, helping behaviors is one of the key components of the Organizational Citizenship Behavior (OCB) construct, and confusingly, within OCB, helping behaviors are considered only those behaviors that are conducted voluntarily [65,66].

In a comprehensive review of prosociality at work, Bolino and Grant [2] differentiated between three distinct concepts that feature in management research:Prosocial motives can be described as the desire to benefit others, or expend effort out of concern for others (which may or may not translate into action). Prosocial motives may be a trait or a state, and may be driven by other-orientated intentions, self-orientated goals, or a mix of both [67].Prosocial behaviors according to Bolino and Grant [2] (p. 602) “are acts that promote or protect the welfare of individuals, groups, or organizations”. These behaviors can be directed at colleagues, clients, teams, stakeholders, or the organization more broadly. These behaviors can be role-prescribed (in-role behaviors) or discretionary (extra-role behaviors).Prosocial impact “refers to the experience of making a positive difference in the lives of others” [2] (p. 602). Although it takes an outcome-focused outlook to organizational prosociality, it relies solely on the actor’s perspective, therefore raising issues around people’s capacity to correctly evaluate the outcomes of their actions.

Interestingly, the authors do not offer an overreaching umbrella term or definition that encompasses all three concepts. We propose that the term organizational prosociality may be more apt than prosocial organizational behavior to represent the three terms overall.

Although within this classification the description of prosocial organizational behaviors may seem deceptively similar to Schroeder and Graziano’s [64] definition, it differs in the sense that it untangles prosocial behaviors from their outcomes, therefore focusing solely on the behavior itself. However, similar to the issue raised above, the difference between prosocial behaviors conducted at work and professional helping behaviors remains vague.

Another point of difference between the conceptualization of prosociality applied in the psychology literature compared to the organizational literature is the tendency to use more narrow conceptualizations in the management literature. For example, the use of the term “motives” in Bolino and Grant’s [2] work is more restricted than Penner et al.’s [27] definition of intraindividual tendencies and dispositions. Similarly, the term “impact” [2] particularly, as it refers solely to the actor’s perspective, is narrower than references made in the psychology literature to outcomes or consequences of prosociality, often with the acknowledgment that they should be evaluated from multiple perspectives [59]. As Bolino and Grant [2] (p. 647) pointed out: “Perceptions of prosocial impact are in the eye of the beholder, and when employees’ judgments diverge from beneficiaries’ and societal perspectives, they may slide down the slippery slope of justifying all manner of sins”.

Importantly, while Penner et al.’s [27] multilevel analysis recognizes macrolevel concepts in which prosociality can occur on a group, community, or organizational level (such as prosocial norms, or corporate social responsibility), Bolino and Grant’s [2] definition of prosocial behavior is more confined to behaviors enacted by individuals, rather than phenomena that exist on a group or organizational level.

The semantic analysis presented here suggests that the definitions of prosociality drawn from psychology slightly differs from those used in the organizational literature. Theoretically, the umbrella term organizational prosociality should comfortably correspond with the broader prosociality umbrella concept; however, the discrepancies detected here between the two disciplines suggest that the meanings that are attached to prosociality indeed differ across disciplines, and that some clarification of the nuanced points of divergence is still required in order to avoid the typical pitfalls of interdisciplinary research [49].

Another point of disagreement emerges with regard to the concept of kindness, which we turn to next.

## 4. Kindness

The scientific work around kindness is currently experiencing an upsurge of interest and publications, both in psychology and within the work domain [68,69]. However, while the concept of prosociality has a history of nearly 40 years of research, the concept of kindness, which is considered a subdomain of prosociality, is still at its early stage of development, which is apparent through some of the discrepancies and contradictions that exist around its definition and theoretical underpinning.

According to Curry et al.’s [5] often-used definition, kindness is an umbrella term that encompasses a range of dispositions and interpersonal behaviors that are intended to benefit others. Puzzlingly, this definition seems to be identical to Batson and Powell’s [57] definition of prosociality cited above. As these terms are used interchangeably at times (see, for example, [50,51,52,53,54,55]), this raises the question of how do prosociality and kindness align and how do they differ?

An additional area of contention is that, similar to prosociality, kindness has multiple definitions in the literature, some of which do not align. For example, within a psychological therapeutic context, Kerr, O’Donnovan, and Pepping [70] (p. 20) defined kindness as “a combination of emotional, behavioral, and motivational components” highlighting its emotional undercurrent—compassion. The authors also argued that it has an altruistic (selfless) motivation. However, compassion and altruism do not seem to feature in definitions of kindness applied in other subdomains of psychology. In a developmental psychology context, Knafo and Israel [71] defined kindness as a constellation of positive dispositions, emotions, and behaviors towards others. In positive psychology, Lyubomirsky, Sheldon, and Schkade [72] conceptualized kindness as a behavior costly to the self that benefits the other. Within a health psychology context, Campling [73] (p. 3) offered the following definition: “Kindness implies the recognition of being of the same nature as others, being of a kind, in kinship. It implies that people are motivated by that recognition to cooperate, to treat others as members of the family, to be generous and thoughtful. The word can be understood at an individual and at a collective level, and from an emotional, cognitive, even political point of view”.

In one of the most extensive analyses currently available of the concept of kindness, from a developmental psychological perspective, Malti [68] conceptualized kindness as an ethical interpersonal virtue, a value, and a behavior characterized by a sincere, deep care for others. As such, it reflects a cognitive and emotional state that engenders sensitivity to the similarities between ourselves and others, and a recognition of people’s uniqueness. It involves extending attention, consideration, and care for others, and the motivation to support and promote their welfare and development. It therefore entails the capacity to expand or transcend our sense of self, and “go beyond self-protection or group promotion and towards a focus on the broader social good” [68] (p. 631).

Malti [68] went on to differentiate between three components of kindness:Kind cognitions involve a variety of traits or states such as perspective taking, open-mindedness, respect, and understanding, which can help us recognize other people’s standpoints, and incorporate these into our own. Taking a reflective, open-minded, and respectful stance toward others, and understanding their perspective and the relativity of other people’s positions, are key components of kindness, since those cognitions highlight our shared humanity and interdependence.Kind emotions entail mainly other-orientated moral emotions, whether state or trait-like, such as sympathy, empathy, compassion, tenderness, or gratitude. They can also include moral emotions that are self-orientated, such as guilt, embarrassment, or shame, which in the context of kindness may occur when one regrets wrongdoing [74]. They differ from other emotions (such as sadness, joy, or calm) since they are linked to ethical values and social norms, and tend to transpire when a person experiences moral judgment [74]. Similar to other emotions, these emotions have action tendencies and can prompt a variety of behaviors.Kind behaviors or actions, often denoted in the literature as acts of kindness, refer to other-orientated behaviors. These behaviors may entail a wide array of behaviors ranging from modest everyday acts of kindness (such as listening, smiling, reassuring, or paying a compliment) to more complex behaviors (such as helping, collaborating, sharing, or supporting).

According to Malti [68], the three components are interconnected with cognitions and emotions prompting each other, and both inducing kind behaviors.

Malti’s [68] analysis suggests that, similar to prosociality, kindness is a multidimensional concept that can manifest as a variety of dispositions—a trait and a state, an intention, a moral emotion, a motivation, an attitude, an interpersonal orientation, and a virtue. It can also manifest in a multitude of behaviors [34,68,75]. This suggests that there are numerous areas of intersection between kindness and its umbrella term prosociality. However, Matli’s [68] conceptualization indicates (though not clearly stated) that acts of kindness and prosocial behaviors differ in their actors, target recipients, and scale. Drawing on Penner et al.’s [27] multilevel typology and on Malti’s [68] analysis, we argue that kindness can be situated at the micro level and on the meso level, hence encompassing intraindividual dispositions and tendencies, as well as interpersonal behaviors or actions that are enacted by an individual and directed at a person or a small group. However, it is less likely to encompass macro level concepts, as these are either enacted by or directed at a much larger entity: an organization, a community, a nation, or society at large. Hence, examples such as a person or an organization giving a donation to a charity, or engaging with pro-environmental activism are considered prosocial behaviors rather than acts of kindness. Additionally, Sanderson and McQuilkin [54] coined the term “everyday kindness” to delineate a class of voluntary, low-cost actions that are intended to promote the well-being of others in everyday situations through small-scale expressions of attention, thoughtfulness, and care. This situates everyday kindness as a distinctive class of prosocial behavior that is different from the more costly actions often included under the umbrella of prosociality [27].

## 5. Kindness within Organizations

Presently, there is scarce literature on the manifestations of kindness in organizational settings [76], and the concept of kindness does not feature in most textbooks on management and organizational studies. Coller [76] maintained that the lack of attention to kindness within the management literature is because it has been viewed as irrelevant and due to the lack of empirical tools and measures that could explain the role of kindness within organizations. However, Waddington [77] argued that we are currently witnessing a “compassion turn” that acknowledges the vital role of basic manifestations of humanity and care in organizations. Hence, more recently kindness as a topic has been receiving some attention, mainly in two domains: leadership [69] and healthcare [78].

Despite the scarcity of research into kindness in organizational settings, within the literature that exists, the concept of kindness lacks clarity, and it is often left undefined [76] or used interchangeably with other terms that fit under its umbrella, such as compassion [79], benevolence [80], altruism [81], and prosociality [51,82]. For example, Allen [83] offered a definition that seems to align with Malti’s [68] definition reviewed earlier: kindness involves being, doing, and feeling, and also has a motivational factor. Similarly, Fryburg [84] noted that the umbrella term of kindness encompasses numerous prosocial emotions and behaviors, including caring, generosity, altruism, empathy, gratitude, and compassion. Coller [76] (p. 16), however, provided a narrower definition that is confined to behaviors, and excludes the multitude of dispositions and tendencies that fall under its remit: “acts by individuals that are relational (i.e., actions to promote change in others) and discretionary (i.e., recognition that others are in need) in nature”. Gibb and Rahman’s [79] (p. 584) focused in their analysis on four dimensions associated with kindness: “kindness associated with an ethics of care; kindness as an interpersonal trait within agreeableness; kindness as reflecting the expectation of reciprocal gain; and kindness as a concomitant of communitarian relations”. Similarly, in a leadership context, Baker and O‘Malley [85] explored the concept of kindness in an attempt to explain what “leading with kindness” means. The authors classified six fundamental features of the construct: compassion, gratitude, integrity, authenticity, humility, and humor. In a similar vein, within the healthcare context, Dossey [86] (p. 358) defined “basic kindness” as the quality of being “friendly, generous, considerate, empathic, and compassionate”.

In view of the diverse definitions, Allen [83] (p. 60) made the following critique: “The etymology of kindness is clear, but the enactment can be fuzzy. There is a gap between theory and praxis of kindness. Acts of kindness can become lost in other behaviors and actions because of the depth and breadth of kindness and its close connection with other concepts”.

In one of the most extensive works that explores kindness in leadership. Caldwell [87] emphasized the merit of kindness as a moral duty of human resource managers, and conceptualized it as an ethically and morally based leadership concept that entails six elements:Authenticity: Kind authentic behaviors are performed by people who are true to themselves and others, and are driven by a person’s intrinsic beliefs, rather than the impression management.Humanity: Kind acts driven by the value of humanity reflect a person’s moral duty to avoid harm to others and create value for the organization and the greater goodRespect: Kindness embodies respect in the sense of interactional justice: treating people as valued partners, with courtesy and appreciation.Perspective: Kindness requires a person to take perspective: Understand other people’s needs and points of view and the context of situations.Integrity: As an aspect of kindness, integrity involves loyalty to others, speaking the truth, keeping to one’s commitments, and honoring one’s promises to others.Competence: The integration of kindness and competence is required to create systems through which kindness can become an aspect of the organizational values and culture.

These examples of some of the conceptions of kindness that feature in management literature suggest that they align in several key aspects with the psychological definition of kindness cited earlier, since they refer to intrapersonal dispositions and interpersonal behaviors that are intended to benefit others. As such, they can be seen as distinct from the definitions reviewed earlier of prosociality, as they emphasize the smaller scale and interpersonal nature of kindness. However, they also vary between them in nuanced details that are rarely unpacked or thoroughly explored, thereby creating significant confusion around the concept [5].

## 6. Conclusions

Definitions matter. Without semantic clarity around terminology, including basic terms, complex constructs, and umbrella terms, researchers risk stumbling on the jingle-jangle fallacies. Kelley [88], who coined the phrase, suggested that a jingle fallacy occurs when a concept fails to distinguish between different constructs, while the jangle fallacy occurs when two concepts with different names in effect refer to the same construct.

The multitude of new concepts that have emerged in the past decade within the prosociality literature indicates that the field is experiencing a renewed interest, in particular within the work and organizational domain. However, as we have demonstrated, researchers still disagree on definitions of concepts, whether basic, complex, or umbrella terms, dissimilar concepts are often used interchangeably, and the relationship between such constructs is rarely clarified. The case of prosocial organizational behavior and professional helping behaviors that we explored, demonstrates this point.

Two of these concepts, prosocial behaviors and kindness, are particularly baffling, as they are defined similarly, often used interchangeably, and researchers rarely clarify how these concepts differ. Consequently, the field suffers from conceptual vagueness, contradictions, and confusion, which hampers its development. To address this conceptual ambiguity, this paper unpacked the two umbrella terms, both as they occur in the psychological literature and in the management domain to clarify how they differ and where their boundaries lie.

Our analysis suggests that the concept of kindness is an umbrella term that is situated within the wider umbrella term of prosociality. Drawing on Bailey et al.’s [3] and on earlier work by Eisenberg et al. [58] and Penner et al. [27], we propose the following nuanced definition of prosociality both for psychology and for business studies:


*“An umbrella term that encompasses dispositions, voluntary behaviors and processes that are focused on or contribute to the welfare of others, and can emerge at three levels: the micro—intrapersonal level, containing mainly dispositions and tendencies, the meso level, which includes behaviors that are enacted by and directed at a small-scale beneficiary (a person or a small group), and macro level, which involves behaviors that are enacted by and directed at large scale recipients (such as organizations or communities) as well as group or organizational processes”.*


In comparison, kindness is much smaller in scale and differs from prosociality in its actors and target recipients. It overlaps the concept of prosociality in the sense that it includes both dispositions and behaviors, and can occur on the micro and meso levels. However, kindness differs from prosociality as it does not encompass the behaviors and processes that occur on a macro level. We therefore propose the use of a more nuanced definition of kindness:


*“An umbrella term within the domain of prosociality, that includes a range of intrapersonal dispositions, and voluntary interpersonal behaviors that are intended to benefit an individual or a small group”.*


The proposed definitions also disentangle prosocial behaviors and kindness from helping behaviors. As noted earlier, the literature distinguishes between professional helping behaviors that are role-prescribed, from voluntary helping behaviors [36] denoting that only helping behaviors that are discretionary are considered prosocial, while those that are driven by professional duties, should not be placed under the umbrella of prosociality. Drawing on Collett and Morrissey [14] and on Bierhoff’s [36] work, we propose that


*“voluntary helping behaviors is a subtype of kindness. It entails voluntary interpersonal interactions where people offer each other some form of support”*


With regard to future work, research on prosociality, kindness, and helping behaviors both in psychology and in the organizational domain would benefit from a review that encompasses the multitude of concepts that each of these entail and clarify their overlaps and distinctions, hence untying the text, the context, and their nuanced subtext.

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
