# Peer review of "Untying the Text: Organizational Prosociality and Kindness"

_behavsci, 2023, doi:10.3390/bs13020186_

Round 1

Reviewer 1 Report

Taken on its own terms this is an insightful and illustrative analysis of the general problem of ambiguity around constructs in use in this domain with reference to two specific constructs. It aims to deal with three contextual causes of that ambiguity- levels of analysis, breadth of concepts, and multiple disciplines. It is mainly centred on the first two rather than than the third in analysis and conclusions. there is scope in the conclusion to revisit the inter-disciplinary context.

I had a number of questions after I read this which might be worth some consideration in revisions, though its not essential.

why has there been an upsurge in interest in the last decade on these themes ? Given the major economic, technological, environmental and social challenges experienced in organizations in the last decade why has prosociality become prominent and more worthy clarification ?

Management can be perceived as a field full of fads, fashions and fundamentals; are there fad and fashion elements to the prosocial, or is the interest in it reconnecting with something more fundamental that has been neglected ?

Is it not possible to do the sifting and sorting of constructs with respect to the extent to which they have been/can be measured ? there is little discussion of the merits of measures, or the measurement process, as a means of settling on a set of constructs which differentiate and align.

Asking what is the opposite can be revealing; what are the opposites of kind and of prosocial ? The opposite of kind may seem simple, but then cant it be cruel to be kind ? And cant kindness become stultifying ?  I don't agree that the opposite of prosocial is antisocial (which is a loaded term with its own problems) but more to do with self-interest and 'me, me, me'.

I find there is a lack of cases or examples. These can be illuminating among the dense description of constructs. I like jingle jangle, but note sure I really got that. 

Are there not generational, inter-cultural and organization contexts which are more important to understand and than find distinctions among aligned constructs. Prosociality as self-lauded and prescribed in collectivist cultures often excludes, for example, LGBT+. Are hard rights then not more important than soft constructs (like kindness and prosociality) for achieving the behaviors desired  ?

Author Response

Please see the attached document with our response.

Reviewer 2 Report

Thank you for the opportunity to review this article addressing commonalities and differences between prosociality and kindness. Below, I provide some ideas for the authors to improve this work.

1) Motivation

The authors establish a clear goal for the paper upfront, “provide a review of the definitions of prosocial behaviour and kindness, in an attempt to semantically untie the text, unpack the context, and discuss the subtext that underlie these concepts.” However, the subsequent discussion seems to fall somewhat short of this goal, leaving it unclear as to why and how this article advances our understanding of these topics beyond prior work.

a.  First, it is unclear that any work has in face “tied” these concepts together in a way that requires the authors to “untie” them. In this realm, the authors acknowledge little research on kindness within organizations, suggesting these concepts may not have previously been strongly linked (or at least linked in a way that complicates understanding of the literature). Notably, the majority of the prosociality reviews the authors mention do not invoke the concept of kindness (e.g., Bolino & Grant, 2016; Penner et al., 2005; and Pfattheicher et al., 2022 do not). The authors provide some definitions that seem to overlap, but these seem to draw from a handful of specific papers. This treatment leaves it unclear as to why this semantic work advances the understanding of the literature. The authors argue that “the field suffers from conceptual vagueness, contradictions and confusion which hampers its development,” yet do not present evidence that such suffering is apparent.

b. To this end, it would be very valuable for the authors to highlight specific examples from the literature which conflate the two topics, and illustrate how and why prior treatment of them has created an issue. The authors do generally discuss how conceptual ambiguity in the prosociality literature has been a problem, but they do not clarify how this relates to kindness specifically, nor how their efforts to disentangle these two constructs differs meaningfully from efforts made by others to categorize the prosociality literature (e.g., Bolino & Grant, 2016; Gilbert et al., 2019; Pfattheicher et al., 2022).

2) Emphasis on disentangling prosocial behavior and kindness versus reviewing past work

a. The article does a good job summarizing the various ways in which prosociality has been considered. However, it does not give substantial treatment to examining differences between prosocial behavior and kindness. The authors discuss how Malti’s (2021) work suggests that kindness does not apply to the organizational domain (which may be due to the different focus of that work, see #3 below), but go on to highlight ambiguity around the kindness construct without comparing it with prosocial behavior further. If the article is intended to separate these two constructs, it would be valuable to do so more thoroughly.

3) Reliance on specific definitions

a. The authors’ arguments about the overlap between prosocial behavior and kindness seem to rely on one definition of kindness from the developmental psychology literature (Malti, 2021), which is not intended to focus on organizational-level actions. This suggests that their conclusion that kindness does not include organizational-level behaviors (the main differentiating point they raise) may be incomplete. If the authors are, in fact, drawing from more extensive work to make this conclusion, it would be valuable to include that.

b. It is also unclear from the definitions offered that kindness would necessarily exempt organizational actions. In this vein, collaborating, sharing, and supporting (elements of the kindness definition) seem that they may be actions that can be undertaken at the organization level, such as, for example, contributing to a knowledge platform (e.g., Wasko & Faraj, 2005).

3) Minor - typos

-“ Pfattheicher et al., 2022 Gilbert, 80 Basran, MacArthur & Kirby, 2019” (missing ;), p.2

-“In an early paper Dovidio (1984, p. 363) noted 84 that the terms prosocial behaviour” (missing comma after “paper”, p.2)
-“Prosiciality”, p.3

Author Response

Please see our notes in the attached document.

Reviewer 3 Report

In my opinion (I deal with economics and management) the article should be supplemented in the part concerning organization (and conclusions) with the following problems:

- prosociality and kindness build the right atmosphere of the organization,

- employees (members of the organization) expect a favorable atmosphere in addition to appropriate working and remuneration conditions,

- prosociality activities and kindness increase the contentment, satisfaction and loyalty of employees (members of the organization), which is beneficial for the organization (and in the long run - profitable).

The above issues can be found in the CSR literature.

Author Response

(The authors gave the same response as above.)
